# Influence of Formulation Factors, Process Parameters, and Selected Quality Attributes on Carvedilol Release from Roller-Compacted Hypromellose-Based Matrix Tablets

**DOI:** 10.3390/pharmaceutics14040876

**Published:** 2022-04-16

**Authors:** Aleša Dular Vovko, Bor Hodžić, Tina Brec, Grega Hudovornik, Franc Vrečer

**Affiliations:** 1Krka, d. d., Novo mesto, Šmarješka cesta 6, 8501 Novo mesto, Slovenia; alesa.dular-vovko@krka.biz (A.D.V.); bor.hodzic@krka.biz (B.H.); tina.brec@krka.biz (T.B.); grega.hudovornik@krka.biz (G.H.); 2Faculty of Pharmacy, University of Ljubljana, Aškerčeva cesta 7, 1000 Ljubljana, Slovenia

**Keywords:** roller compaction, hypromellose, matrix tablets, drug release, spatial filtering technique

## Abstract

The importance of roller compaction is recently increasing. This study evaluates the combined effects of formulation factors, process parameters, and selected quality attributes on drug release from roller-compacted hypromellose-based matrix tablets containing carvedilol as a model drug. The influence of selected factors was statistically assessed and good predictive models were developed for various time points of the release profile. The results show that the release profile is mostly affected by the particle size distribution of granules and roll speed. This indicates that the roller compaction process has a major impact on drug release, which is also formulation dependent. A higher d50 and lower d90 value of spatial filtering technique-based particle size distribution results, a lower roll speed, increased hypromellose content, using microcrystalline cellulose as a filler, and higher tablet hardness, resulted in a decrease in the drug release rate. On the other hand, the effect of the roll pressure, size of screen apertures, and d10 values on drug release was insignificant. The significance of the factors was further explained by granule shape, their porosity, and friability evaluation, and by compressibility and compactibility studies of compression mixtures. Additionally, the spatial filtering technique demonstrated to be a promising tool in controlling the roller compaction process.

## 1. Introduction

Hydrophilic matrix tablets for extended drug release are widespread dosage forms because they offer benefits for patients, such as improved compliance and reduced adverse effects [1,2,3,4,5]. Moreover, hydrophilic matrix-forming polymers have a low cost, offer good compatibility with a broad spectrum of APIs (active pharmaceutical ingredients) and excipients, and are compatible with conventional manufacturing steps for tablet manufacturing [6,7,8]. One of the most widely used hydrophilic polymers in matrix systems is hypromellose (hydroxypropyl methylcellulose (HPMC)). For manufacturing hypromellose tablets, simple technology, such as direct compression may be used [3,6]; however, poor flowability of compression mixtures with high hypromellose content may cause manufacturing problems, such as mass variations of compressed tablets and segregation, resulting in content uniformity issues. To overcome these problems, granulation may be applied as a manufacturing technology. Wet granulation is commonly used; nevertheless, it can be challenging because hard lumps and unwetted areas may occur and cause non-homogeneous wetting and swelling of hypromellose [9,10], which may result in variable dissolution profiles. Dry granulation by roller compaction is an alternative approach that can eliminate such problems. Moreover, this technology is simple, cost-effective, environmentally friendly, and suitable for heat- and moisture-sensitive materials. In addition, the scale-up and post-approval process changes are easy because this is a continuous process [11,12]. On the other hand, there are some limitations to roller compaction, such as the possible excessive amount of fines due to the absence of liquid binder and reduced tensile strength of tablets as a result of re-compaction of the materials [12,13,14]. Additionally, roller compaction is not suitable for materials that are susceptible to mechanical stresses, since compression during processing may lead to phase transformation via the solid-state or melt mechanisms [15]. In some cases, mechanical stress exposed to the drug particles during compaction may also induce chemical degradation of the drug. However, it is, in most cases, possible to overcome the challenges of roller compaction with a selection of appropriate equipment and by technological approaches to maintaining quality during the process [14,16].

Although roller compaction is becoming the preferred granulation method in the pharmaceutical industry [12,17,18], only a few studies evaluating dry granulation for manufacturing hypromellose matrix tablets are available. Hariharan et al. [19] examined the effects of selected formulation variables (the content of hypromellose, MCC (microcrystalline cellulose) and glyceryl dibehenate) on the physical properties of tablets and the release rate of diclofenac from extended-release tablets prepared with roller compaction. They developed a model for drug release at t70 (time after 70% of the drug has been released) and demonstrated that the drug release was slower for formulations containing a larger amount of hypromellose and glyceryl dibehenate and faster for a larger amount of MCC. Sheskey and Hendren [20] studied the influence of the selected process variables—roll speed, feed screw speed, and roll pressure equipment—and the hypromellose content on the release of theophylline. The hypromellose content had a significant effect on drug release, whereas the process variables and compression force used for manufacturing the tablets did not alter the drug release. Various studies [19,20,21] report a lower crushing strength of the tablets due to a reduced compactibility of the materials [22,23,24], which implies that roller compaction may not be suitable for hypromellose-based tablets. Heiman et al. [25], who studied the combined effects of roller compaction process parameters and composition on the manufacturability of tablets containing hypromellose matrices, addressed this issue. They confirmed that roller compaction was suitable for brittle and plastic deforming drug substances with various particle sizes; however, they could not confirm the effects of the process parameters on drug release.

The aforementioned studies mainly focused on the manufacturability of roller compaction and only partially evaluated the different effects on drug release from hypromellose-based tablets. The aim of this study was to provide a comprehensive and systematic approach to understanding the combined effects of formulation, process variables, and selected quality attributes on drug release from hypromellose-based hydrophilic matrix tablets. Regarding formulation variables, tablets contained various hypromellose content and two different fillers: lactose, which predominantly deforms by fragmentation, and MCC, a plastically deforming material. Roll pressure, roll speed, and the screen aperture size of the oscillating sieving device were selected as the process variables in this study. In addition, the effect of the particle size distribution (PSD) of the granulate and tablet hardness on the drug release rate was also statistically evaluated within the study. The selected parameters that proved a statistically significant effect on the drug release rate (roll speed, PSD of the granulate) were further investigated by granule friability, shape, and porosity evaluation, and by compressibility and compactibility studies of compression mixtures of selected batches to improve understanding and confirm the significance of the factors.

## 2. Materials and Methods

### 2.1. Raw Materials

Carvedilol (provided by Krka, d.d., Novo mesto, Slovenia) is a beta-blocker and was used as a model drug. The strengths of carvedilol tablets for immediate release on the market are 3.125 mg, 6.25 mg, 12.5 mg, and 25 mg, which should be administered twice daily. In this study, prolonged-release matrix tablets were developed, containing 25 mg of carvedilol, which would be suitable for administration once daily. Hypromellose (METHOCELTM, K4M Premium CR grade, Colorcon, Dow Chemical Co., Midland, MI, USA) was used as the matrix-forming polymer. Lactose monohydrate 200 mesh (DFE Pharma, Goch, Germany) and MCC (CEOLUSTM KG802, Asahi Kasei Corporation, Tokyo, Japan) were utilized as fillers in this study. The tablets also contain magnesium stearate (Faci SpA, Carasco, Italy) and colloidal silicon dioxide (AEROSIL^®^ 200 Pharma, Evonik Degussa GmbH, Essen, Germany).

### 2.2. Composition of Tablet Formulation Tested

The tablets evaluated in this study were composed of 15.0% carvedilol (added intragranularly), 25.0% hypromellose (added intragranularly), 58.4% filler (added intragranularly)—lactose monohydrate (incorporated in samples no. RL-XX and TL-XX) or MCC (incorporated in samples no. RM-XX and TM-XX)—1.3% magnesium stearate (0.65% added intragranularly and 0.65% added extragranularly), and 0.3% colloidal silicon dioxide (added extragranularly). In the selected experiments, a larger amount of hypromellose was compensated for with filler; hence, the tablets contained 35.0% hypromellose and 48.4% filler, or 45.0% hypromellose and 38.4% filler, respectively.

### 2.3. Compression Mixture Preparation and Tableting

The composition and process of producing the granulate were the same as in our previous study [26]. Key information on the formulation and process parameters of the batches that were utilized in this study is presented in Table 1.

Magnesium stearate and colloidal silicon dioxide were added to the granulate and blended for 1 min at 32 rpm in a 6 L bin mixer to prepare the compression mixture. Prior to mixing, magnesium stearate and colloidal silicon dioxide were sieved manually through sieves with 0.6 mm apertures. The batch size was approximately 1 kg.

The compression mixture was compressed into tablets using a rotary tablet press Pressima (IMA Killian GmbH & Co. KG, Cologne, Germany) running at 10 revolutions/min using biconvex oval punches (11 mm × 5.5 mm). The main compression and fill depth were minimally adjusted between different mixtures to achieve the target average weight of the tablets: 166.6 mg and two target average hardnesses of the tablets: 80 N and 100 N.

### 2.4. Data Acquisition and Data Analysis

#### 2.4.1. Spatial Filtering Technique

At-line particle size measurement was performed with the spatial filtering technique (SFT). The methodology and the results are presented in our previous article [26]. The d10, d50, and d90 results of volume-based PSD were utilized in the statistical analysis.

#### 2.4.2. Tablet Weight and Hardness Evaluation

The weight and hardness of tablets were measured with a fully automated tablet tester Multicheck (ERWEKA GmbH, Heusenstamm, Germany) by utilizing a constant speed measuring principle. The result was given as an average of 20 measurements.

#### 2.4.3. Drug Release Testing

To obtain drug release profiles, a dissolution apparatus with a paddle method (apparatus 2, Agilent 708-DS, Agilent Technologies, Santa Clara, CA, USA) and Japanese sinkers were used in 900 mL of acetate buffer solution (pH = 4.5) at 37 °C ± 0.5 °C with a paddle speed of 100 rpm. The selected method was shown to be the most discriminatory in previous studies [27]. The amount of drug released was determined spectrophotometrically from the absorbance measured at 285 nm at 25 preselected time points. The sampling during the analysis was carried out automatically by using flow-through cuvettes. The % of the drug released was calculated using a pre-prepared calibration curve. Preselected drug release profiles are presented in the article to demonstrate how significant factors affect the release rate. The similarity factor (*f*2) was also calculated for the comparison of dissolution profiles.
(1)$$f2=50\times \log\;\{[1+\left(\frac{1}{m}\right)50\times \log\;\{[1+{\left(1/m\sum_{t=1}^{m}{\left(Rt-Tt\right)}^{2}\right]}^{-0.5}\times 100\},$$

In Equation (1), m represents the number of time points, Rt is the dissolution value of the reference batch at time *t*, and *Tt* is the dissolution value of the test batch at time *t* [28]. In the calculation of *f*2, only one time point above 85% of the released drug has been considered.

To evaluate the kinetics and drug release mechanism, the power law [29] model was used:(2)$$\frac{Mt}{M\infty }=k*t^{n},$$
where *Mt* represents the amount of drug released at time *t*, *M*∞ is the amount of drug released after infinite time, *k* is a release rate constant and n is the diffusional exponent that characterizes the release process.

Where relevant, the diffusional exponent of the power law model and *f*2 factors among profiles of the manufactured batches are mentioned in the Section 3, but they are not fully presented in the article.

#### 2.4.4. Statistical Data Analysis

Multivariate data analysis was performed using Unscrambler^®^ 11.0 (Camo Analytics AS, Oslo, Norway) multivariate data analysis software package, including Design-Expert^®^ v12.0.3.0 (Stat-Ease, Inc., Minneapolis, MN, USA). The effect of the selected factors on the drug release rate has been evaluated with statistical analysis of the 25 aforementioned batches prepared with different compositions, process parameters, and quality attributes. The independent variables/factors investigated were the following: the content of hypromellose, filler type, roll speed, roll pressure, screen aperture size, PSD of granules (d10, d50, and d90), and tablet hardness. Data were inversely transformed and assessed by multi-way analysis of variance. Reduced linear predictive models were developed for the following time points: 1, 2, 3, 4, 5, 6, 7, and 8 h.

To further investigate and confirm the effects of the significant factors, the following methods (see Section 2.4.5, Section 2.4.6, Section 2.4.7, Section 2.4.8) were utilized for the analysis of selected granulate and compression mixture batches.

#### 2.4.5. Laser Diffraction

The laser diffraction method was performed using a Malvern Mastersizer 2000 (Mastersizer 2000, Malvern Instruments Ltd., Worcestershire, United Kingdom) with a Scirocco 2000 automated dry powder dispersion unit. The laser diffraction (LD) method was utilized for granule friability evaluation. The measurements were carried out at pressures of 1 bar and 3 bars. The friability of the selected batches of granules was calculated from the difference (in %) between the measurements at pressures 1 bar and 3 bars for d [4, 3] and d90 values.

#### 2.4.6. Dynamic Image Analysis

The sphericity and aspect ratio of selected samples of granules was determined by dynamic image analysis (DIA) using a Camsizer Image analyzer (Camsizer XT, Retsch Technology GmbH, Haan, Germany) by utilizing an x-Dry dispersing unit and x-Jet measuring cell.

#### 2.4.7. True and Bulk Density

True density was determined by a helium pycnometer (AccuPyc II 1340e apparatus; Micromeritics, Norcross, GA, USA). The results were given as the average of 10 measurements. Determination of bulk density was performed according to the method described in Ph. Eur. (10th ed., 2.9.34.). One repetition of one average sample was performed. Using the results of the true density and bulk density of the granules, the porosity of granules was calculated. The results of true density were also used in compressibility and compactibility studies.

#### 2.4.8. Compressibility and Compactibility Studies

The selected compression mixtures were compacted on a Gamlen D500 compaction simulator (United Kingdom). Tablets of approximately 100 mg of compression mixture were compacted using 6.0 mm diameter flat-faced punches using compaction pressures of 69, 121, and 172 MPa. The compression speed was 10 mm/min and the hold time was 1 s. After compression, the weight (*w*), thickness (*h*), diameter (*D*), and crushing force (*F*; hardness, measured at 80%) of the compacts was determined by using a Gamlen tablet tensile analyser (United Kingdom). The solid fraction (*SF*; see Equation (3)) and tensile strength (TS; see Equation (4)) of the compacts [30] have been calculated in order to obtain compactibility and compressibility plots.
(3)$$SF=\frac{\rho_{app}}{\rho_{t}}=\frac{4w}{\rho_{t}\pi hD^{2}},$$
where *ρ_app_* represents the apparent density of the compacts and *ρ_t_* represents the true density of the material.
(4)$$TS=\frac{2F}{\pi Dh}.$$

The results represent the average of three measurements.

## 3. Results and Discussion

The results of drug release at eight selected time points (1–8 h) were included in the statistical analysis and models have been developed for each selected time point. These time points have been selected in order to thoroughly investigate the part of the release profile where the differences between the examined batches are the most pronounced. Table 2 represents the R^2^, adjusted R^2^ (a measure of the amount of variation around the mean, adjusted for the number of terms in the model), and predictive R^2^ (a measure of the amount of variation in new data explained by the model) of models. The models’ F-values (see Table 2) imply that the models are significant. For the models developed for all time points, there is only a 0.01% chance that the F-value is largely due to noise. High R^2^ values further show that the developed models have good predictive value. Furthermore, *p*-values (see Table 2) less than 0.0500 indicate that the content of hypromellose, filler type, roll speed, d50, d90, and tablet hardness are significant model terms for all the selected time points. Figure 1 further shows the values of coded coefficients for significant factors for each time point.

The results demonstrate that higher hypromellose content, higher tablet hardness and d50 values, and lactose as a filler decrease the release rate, whereas a higher d90 value and roll speed, having a negative value of coded coefficients, cause a faster release of the drug from the matrix tablets. On the other hand, the effect of the d10 value of PSD, roll pressure, and screen aperture size are insignificant.

According to the statistical analysis, initially, the rate of drug release is mainly controlled by the PSD of the granulate. The effect of the d50 and d90 values on drug release rate proves to be greater than the effect of hypromellose content, where a large impact was expected. This is in line with previous studies of hypromellose matrix tablets obtained by roller compaction [19,20] and other technological procedures [31,32]. Namely, a higher concentration of the polymer increases the thickness and viscosity of the gel layer formed upon hydration of the tablet matrix with an aqueous solution. This results in a longer diffusional path for drug release and slower erosion of the polymer chains from the surface of the matrix tablet, which reduces the drug release rate. Moreover, it also alters the drug release kinetics. Furthermore, 80 N tablets containing 25% of hypromellose exhibited a diffusional exponent (n) up to 0.5 (R^2^ ≥ 0.99), which indicates that carvedilol is released from the tablets predominantly by diffusion. In tablets with 35% and 45% hypromellose content the n value was higher (n(RM-10) = 0.74, n(RL-10) = 0.66, n(RL-11) = 0.78)), which suggests that drug release mechanism for batches with higher hypromellose content was controlled by diffusion and erosion for formulations of both fillers. Along with the increasing amount of hypromellose, the amount of filler in the tablet has been decreased. The *f*2 value between the 80 N tablets containing 58.4% of filler (25% hypromellose formulation) and 38.4% of filler (45% hypromellose formulation), was 22.7 for lactose and 36.5 for MCC batches, respectively. In both cases, a larger amount of filler resulted in significantly faster drug release, consistent with the results of previous studies [6,33]. The lower *f*2 value of lactose batches implies that lower filler concentration has a slightly larger contribution to the decrease in release rate than for MCC batches. Namely, lactose is a water-soluble filler that facilitates water diffusion, therefore a higher content of lactose also contributes more to an increased drug release rate than in the case with insoluble filler MCC. However, the release mechanism between the fillers containing the same amount of hypromellose is comparable.

Nonetheless, the results of the PSD effects are not straightforward because higher d50 values decrease the drug release rate, and higher d90 values increase it. In order to obtain a better understanding of this phenomenon, batches with the most expressed differences in the PSD results of both fillers (RM-01, RM-05, and RM-06 for MCC batches, and RL-01, RL-05, and RL-06 for lactose batches) were further examined by granule friability, porosity, and shape evaluation and by compressibility and compactibility studies of compression mixtures. Within the lactose batches (see Figure 2), the slowest release profile was obtained for tablets prepared from granules of batch RL-06 with the lowest d90 value. We believe that the absence of large granules allowed for a better particle rearrangement and greater densification of granules during compression. Thus, the tablets of batch RL-06 exhibited lower porosity and decreased drug release rate.

This phenomenon has also been confirmed by the study of compaction behavior, in which the compressibility of sample RL-06 was the greatest among the batches examined (see Figure 3). Moreover, a compactibility plot (see Figure 4) shows that the solid fraction in the batch RL-06 is the greatest at the given tensile strength, which further confirms that the porosity of tablets with the same hardness is the lowest. In contrast, compressibility was the lowest and the solid fraction at the given tensile strength was the highest for batch RL-01 (the reference lactose batch). As a result, the tablet porosity was the greatest and drug release was the fastest and significantly different in comparison to the profile of the RL-06 batch for 80 N tablets (*f*2 = 34.2). Namely, batch RL-01 had medium d50 and d90 values (see Table 3), which contributed to a faster release rate. The presence of larger granules allowed the formation of new unlubricated areas due to their fragmentation and allowed the formation of new interparticulate bonds. However, granules of batch RL-01 had lower friability (see Table 3) and could therefore be deformed to a lesser extent, which also contributed to greater tablet porosity. On the other hand, the greatest d50 value was obtained by batch RL-07 (see Table 3). Larger particles have limited rearrangement ability, which is also shown in a compressibility plot (see Figure 3), where the slope curve becomes flatter at higher compression forces. Nonetheless, the granules’ friability of sample RL-07 is higher (see Table 3) and granules are more prone to fragmentation during compression. Consequently, smaller particles with unlubricated surfaces can form strong interparticulate bonds, and these result in a lower porosity of the tablets and drug release rate.

The MCC batches examined have a smaller particle size and showed smaller differences in PSD than the lactose batches, and therefore the differences in the drug release rate between the batches with different PSD are also less expressed (the *f*2 values between profiles of 80 N tablets were below 50). The largest granules with the highest d50 value (RM-07) and the smallest granules with the smallest d90 value (RM-06) exhibit similar compressibility and compactibility (see Figure 5 and Figure 6).

Larger granules of batch RM-07 show higher friability (see Table 3) and presumably deform to a greater extent, forming small particles, which can rearrange better upon compression into tablets. Similarly, smaller granules (RM-06) with lower friability (see Table 3), which are less prone to deformation, also have a good ability to rearrange during tableting; hence, the compressibility of batches with the largest and smallest particles is comparable. When the particles rearrange and are in proximity, the formation of interparticulate bonds is facilitated due to the good compactibility of MCC, and therefore the compactibility of the batches with the largest and smallest granules are also similar. On the other hand, granules of batch RM-01 with medium d50 and d90 values exhibited medium friability and higher sphericity (see Table 3). This resulted in a poorer rearrangement ability during compression, and therefore the corresponding compression mixture is less compressible than the batches RM-06 and RM-07 with smaller and larger granules (see Figure 5). Granules of batch RM-01 are less deformed upon compaction and exhibit a lower solid fraction at the given tensile strength (see Figure 6), hence, in comparison to the batches RM-06 and RM-07, the tablets with comparable hardness had greater porosity and the fastest drug release rate.

Aside from being an independent variable, PSD is also a dependent variable, since it is strongly affected by process parameters, especially roll speed, as demonstrated in our previous study [26]. In addition to the indirect impact via PSD, the statistical analysis showed that roll speed also significantly affects the drug release rate. Batches produced with the lowest roll speed (3 rpm) had the lowest drug release profile for both lactose (RL-04) and MCC formulations (RM-04, see Figure 7).

Lower roll speed results in a longer compaction time and a reduced amount of air among the particles in the ribbons. Hence, particles are better rearranged and deformed to a greater extent within the ribbons, which are consequently more mechanically resistant during milling. As a result, granules with lower porosity and larger particle size are produced (see Table 3). They provide more resistance to water diffusion into the granules, resulting in slower drug release. Moreover, the corresponding compression mixtures had the lowest compactibility (see Figure 4 and Figure 6) as a result of loss of compactibility upon roller compaction, significant especially for plastically deforming materials such as MCC [24,34,35]. In the case of lactose, the lower friability of granules allowed good particle rearrangement and prevented the crushing of particles at the early stages of compression. As a result, the tablets with the same hardness had the lowest porosity, additionally resulting in a decreased drug release rate. On the other hand, the batch manufactured at the highest roll speed (7 rpm) for MCC (RM-05) exhibited the fastest drug release (see Figure 7), which differed significantly from the release profile of the batch manufactured at the lowest roll speed (*f*2 (RM-05:RM-04, 80 N tablets) = 48.3). The porosity of these granules was the highest, hence resistance to water diffusion into the granules was the least. In addition, the compressibility, and compactibility of batch RM-05 were the greatest (see Figure 5 and Figure 6) in comparison with batches prepared at lower roll speeds (RM-01 and RM-04). Due to the lowest compaction time, the loss of compactibility was the least, and the porosity of the tablets having comparable hardness was the greatest for this batch, hence the release rate was the fastest. In contrast, lactose batch RL-05, manufactured with the highest roll speed, had a slower release rate. A possible reason is the fragmenting behavior of lactose, for which compaction time does not affect the compactibility, as in the case of plastically deforming filler MCC. Smaller granules obtained by the highest roll speed have a relatively greater part of lubricated surfaces compared to large granules, and thus, the formation of interparticulate bonds is limited. Moreover, the smallest particles could be rearranged better, causing the lower porosity of tablets and decreased drug release rate. Within lactose batches, the drug release profile for the tablet sample based on granulate RL-01 prepared with a medium roll speed (5 rpm) was higher than in the case of batches with lower and higher roll speeds. Namely, the medium granule friability of batch RL-01 allowed fragmentation of granules with the formation of unlubricated surfaces and facilitated interparticulate bond formation. In addition, the granules were more irregularly shaped (see Table 3), which allowed the formation of mechanical interparticulate bonds. Therefore, the solid fraction at the given tensile strength was the lowest, the porosity of the tablets was the highest, and the drug release rate from batch RL-01 was the fastest. In the case of MCC, the batch manufactured at a medium roll speed (RM-01), in comparison with the batches prepared with the highest (RM-05) and lowest roll speeds (RM-04), exhibited a medium granule porosity and granule size, resulting in medium compactibility, medium tablet porosity, and a medium drug release rate.

The effect of roll speed as well as PSD is greater at the beginning of the drug release profile (see Figure 1). The influence diminishes with time, especially after 6 h, when the coherent hydrogel had already formed, and the majority of the drug had been released. Similarly, the influence of hypromellose content decreases over time, presumably because the gel strength is reduced over time due to the erosion of polymer molecules from the surface of the tablets, as demonstrated by Goldoozian et al. [36].

On the contrary, the effect of filler type on drug release is constant through the entire profile, which is expected due to differences in physical properties between both fillers. Lactose is water-soluble and dissolves in the water within the gel layer. This causes the formation of pores in the hydrogel structure and allows faster diffusion of the dissolved drug and accelerated polymer erosion from the gelled matrix. As a result, the drug release rate is increased. On the other hand, MCC is nonsoluble in water, therefore an increase in the release due to filler dissolution in the medium cannot be achieved. According to the statistical analysis (see Table 2 and Figure 1), the drug release rate from the lactose batches proved to be faster in comparison to the sample of tablets containing MCC as a filler (the *f*2 factor value between the reference batches of both fillers, RM-01 and RL-01, is 37.5). Nevertheless, the effect of filler type is less pronounced; that is, the coded coefficients values of the models are lower (see Figure 1) compared to the effect of the hypromellose content, PSD of the granulate, and roll speed. However, its effect is indirectly included in the PSD factor because granules with lactose proved to be larger than MCC granules, as discussed in our previous article [26].

The tablet hardness also influenced the drug release rate, although to a lesser extent. Statistically developed models show that higher tablet hardness results in a lower drug release profile. To obtain tablets with higher hardness, higher compression forces were utilized, which resulted in stronger interparticulate bonding. For lactose formulations, this was made possible by new particle surfaces as a result of fragmentation of the filler particles within the granules. For MCC formulations, plastic deformation of granules occurred to a greater extent when using higher compression forces, which also facilitated bond formation. As a result, tablets with higher hardness expressed lower porosity. This has also been confirmed by compressibility studies, in which the solid fraction increases with compression pressure for all the batches studied (see Figure 3 and Figure 5). Due to lower tablet porosity, the diffusion of water into the tablet’s matrix is slower, hence the drug release is decreased. However, the difference between the drug release profiles among tablets having hardnesses of 80 N and 100 N was not significant (*f*2 < 50) for most of the batches examined, confirming a relatively small effect of tablet hardness compared to other factors. The selection of the hardness of tablets proved to be suboptimal; more tablet samples with more pronounced differences in hardness should be included in the study.

## 4. Conclusions

The drug release rate of carvedilol from hypromellose-based matrix tablets prepared with roller compaction proves to be affected by a combination of formulation, process parameters, and quality attributes. Process parameters, especially compaction time (roll speed) exhibited a major influence on the drug release rate. It proved to affect the particle size distribution, friability, and porosity of granules, which crucially influenced water uptake and the diffusion of the drug from the hydrogel. However, the effects proved to be formulation dependent due to different mechanical deformation mechanisms of MCC and lactose. The effect of tablet hardness was less, whereas screen aperture size, roll pressure, and d10 did not have a significant effect on drug release.

Among the factors examined, surprisingly the d50 and d90 values obtained by SFT have the greatest effect, albeit a contradictory one, on drug release. The effect of PSD is even larger than the influence of hypromellose content. Lower d90 values resulted in better particle rearrangement, lower tablet porosity, and a slower release rate. On the other hand, granules with higher d50 are less hard and tend to crush into smaller particles, which also allows good rearrangement upon compression, and so the release rate also decreases. Therefore, the results indicate that besides the median particle size of the granulate, the amount of large particles obtained by the roller compaction process should also be considered when manufacturing hypromellose-based matrix tablets. Additionally, these results demonstrate that SFT measurements of PSD, together with significant process parameters and a selected formulation, represent a promising tool for monitoring and controlling the production of hypromellose-based matrix tablets prepared by roller compaction. However, the predictive models developed will have to be tested in real-time and on a larger scale, which should be a focus of further studies.

## Figures and Tables

**Figure 1 pharmaceutics-14-00876-f001:**
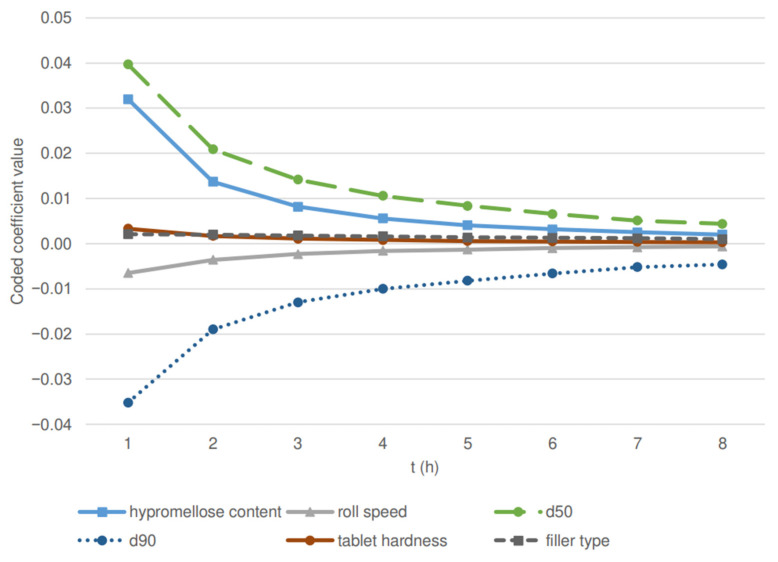
Time dependence of coded coefficients of predictive models for significant factors (hypromellose content, filler type, roll speed, d50, d90, and tablet hardness).

**Figure 2 pharmaceutics-14-00876-f002:**
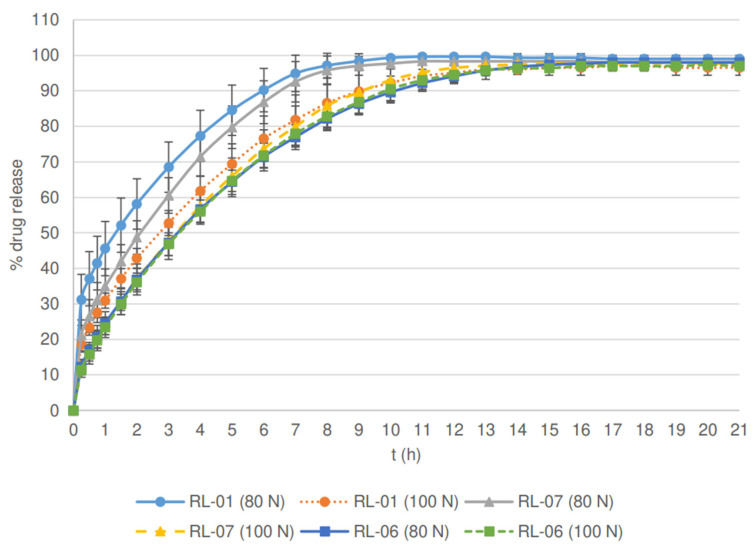
Effect of particle size distribution and tablet hardness on % drug release from matrix tablets containing lactose as a filler.

**Figure 3 pharmaceutics-14-00876-f003:**
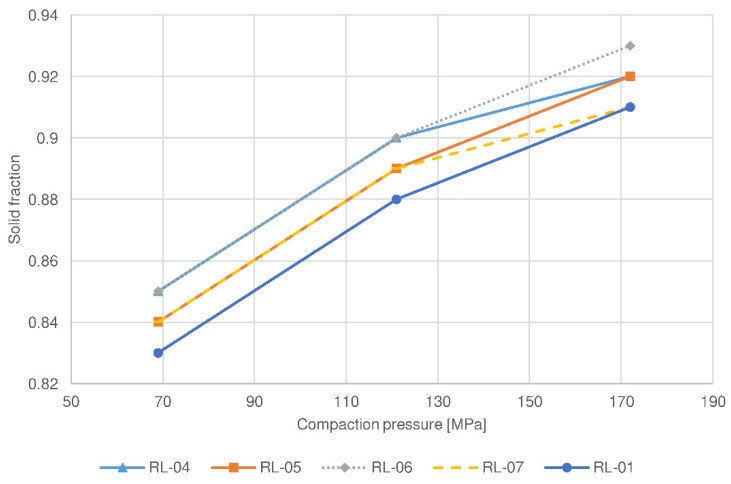
Compressibility plot of selected compression mixtures containing lactose as a filler.

**Figure 4 pharmaceutics-14-00876-f004:**
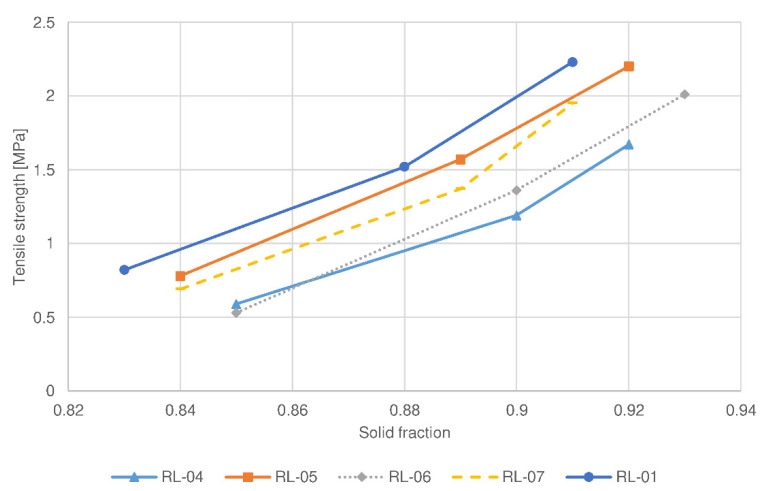
Compactibility plot of selected compression mixtures containing lactose as a filler.

**Figure 5 pharmaceutics-14-00876-f005:**
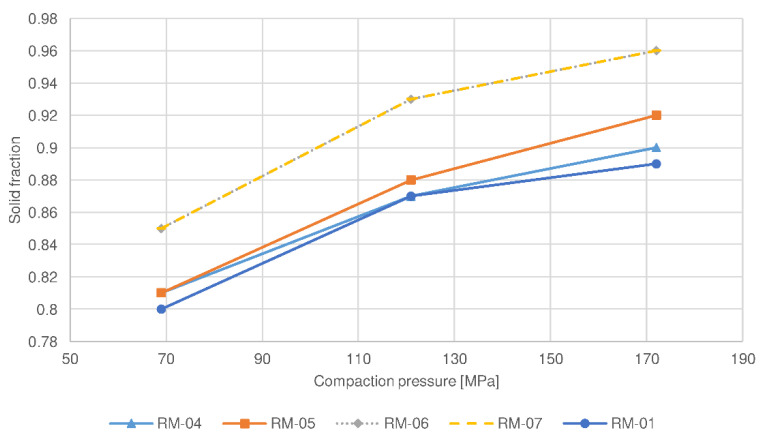
Compressibility plot of selected compression mixtures containing microcrystalline cellulose (MCC) as a filler.

**Figure 6 pharmaceutics-14-00876-f006:**
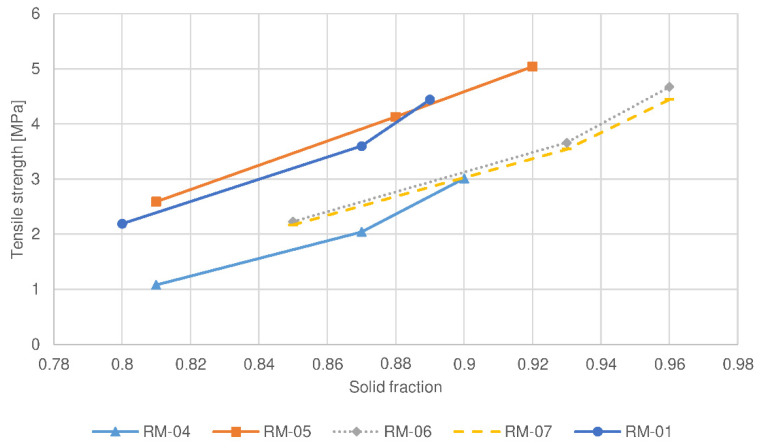
Compactibility plot of selected compression mixtures containing microcrystalline cellulose (MCC) as a filler.

**Figure 7 pharmaceutics-14-00876-f007:**
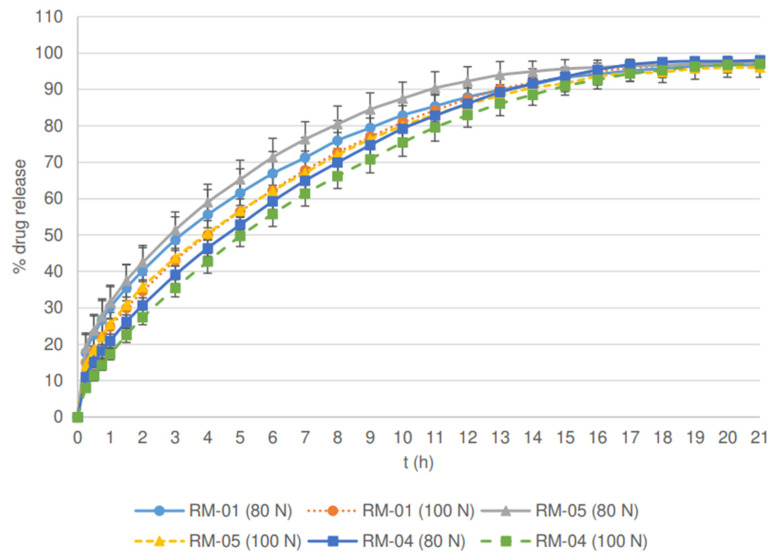
Effect of roll speed and tablet hardness on % drug release from matrix tablets containing MCC as a filler.

**Table 1 pharmaceutics-14-00876-t001:** Process parameters of batches utilized in the study.

Batch No.	Roll Force	Roll Speed	Screen Aperture Size
RL-01 *, RL-12 *, RL-10 **, RL-11 ***, RM-01 *, RM-11 *, RM-10 ***	70 bar	5 rpm	1.00 mm
RL-02 *, RM-02 *	30 bar	5 rpm	1.00 mm
RL-03 *, RM-03 *	110 bar	5 rpm	1.00 mm
RL-04 *, RL-04 *	70 bar	3 rpm	1.00 mm
RL-05 *, RM-05 *	70 bar	7 rpm	1.00 mm
RL-06 *, RM-06 *	70 bar	5 rpm	0.80 mm
RL-07 *, RM-07 *	70 bar	5 rpm	1.25 mm
TL-01 *, TM-01 *	70 bar	5 rpm	1.00 mm
TL-03 *, TM-03 *	100 bar	3 rpm	1.00 mm
TL-04 *, TM-04 *	50 bar	7 rpm	1.00 mm

* 25.0% hypromellose formulation, ** 35.0% hypromellose formulation, *** 45.0% hypromellose formulation.

**Table 2 pharmaceutics-14-00876-t002:** F-value and R^2^ values for predictive models developed at selected time points and *p*-values of significant factors.

Parameter	1 h	2 h	3 h	4 h	5 h	6 h	7 h	8 h
Model F-value	70.27	57.66	52.44	51.02	47.86	46.97	47.54	47.08
R^2^	0.89	0.87	0.86	0.85	0.85	0.84	0.85	0.84
Adjusted R^2^	0.88	0.85	0.84	0.84	0.83	0.83	0.83	0.83
Predicted R^2^	0.86	0.83	0.82	0.81	0.80	0.80	0.80	0.80
*p*-values								
Hypromellose content	<0.0001	<0.0001	<0.0001	<0.0001	<0.0001	<0.0001	<0.0001	<0.0001
Roll speed	0.0027	0.0008	0.0010	0.0011	0.0017	0.0028	0.0025	0.0033
d50	0.0301	0.0206	0.0161	0.0135	0.0132	0.0162	0.0245	0.0200
d90	0.0157	0.0084	0.0057	0.0036	0.0026	0.0029	0.0038	0.0024
Tablet hardness	0.0004	0.0002	0.0004	0.0003	0.0007	0.0012	0.0014	0.0028
Filler type	0.0451	0.0002	<0.0001	<0.0001	<0.0001	<0.0001	<0.0001	<0.0001

**Table 3 pharmaceutics-14-00876-t003:** Results of particle size distribution (d50 and d90 values obtained by SFT*), granule friability calculated from d [4, 3] and d90 values of laser diffraction, granule porosity, and granule shape (sphericity and aspect ratio) of selected batches.

Parameter	RL-01	RL-04	RL-05	RL-06	RL-07	RM-01	RM-04	RM-05	RM-06	RM-07
Filler type	lactose	lactose	lactose	lactose	lactose	MCC **	MCC	MCC	MCC	MCC
Roll speed (rpm)	5	3	7	5	5	5	3	7	5	5
Screen aperture size (mm)	1.00	1.00	1.00	0.80	1.25	1.00	1.00	1.00	0.80	1.25
d50 (SFT *) (µm)	695	715	252	500	877	378	718	113	151	659
d90 (SFT *) (µm)	1153	1392	854	959	1586	907	1248	772	690	1254
Granule porosity (%)	61	59	66	61	59	74	73	79	77	75
Granule friability (LD *** d [4, 3], %)	44	28	53	51	82	36	37	11	35	56
Granule friability (LD d90, %)	46	18	39	55	83	38	27	7	27	68
Sphericity (DIA ****)	0.74	0.73	0.77	0.75	0.72	0.70	0.68	0.70	0.69	0.68
Aspect ratio (DIA)	0.66	0.65	0.66	0.66	0.66	0.58	0.60	0.56	0.57	0.58

* spatial filtering technique, ** microcrystalline cellulose, *** laser diffraction, **** dynamic image analysis.

## Data Availability

Not applicable.

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
