# Peer review of "Influence of Formulation Factors, Process Parameters, and Selected Quality Attributes on Carvedilol Release from Roller-Compacted Hypromellose-Based Matrix Tablets"

_pharmaceutics, 2022, doi:10.3390/pharmaceutics14040876_

Round 1
Reviewer 1 Report
The manuscript entitled “Influence of formulation factors, process parameters, and selected quality attributes on carvedilol release from roller-compacted hypromellose-based matrix tablets” presents a very interesting evaluation of the roll-compaction process. The research plan is well constructed, used methodology is fine and the conclusions are fully supported by the results. There are only few minor issues which should be addressed prior to publication which I listed below, however this does not detract the value of the work presented.
The authors have highlighted the advantages of roll-compaction but they also should comment on the limitations of the technology in the introduction section.
In the methods section the authors describe particle size analysis laser diffraction method as a way to evaluate granules hardness, it rather describes the friability then hardness – it should be changed.
The data for f2 factor calculation should be limited to only one time point above 85% amount of dissolved drug. As in the presented dissolution profiles there are many points above that limit was it calculated properly?
In the figures 2 and 7 there is no need to use two decimal places in Y-axis
In overall the manuscript is well written and legible and it can be accepted after minor revision.
Reviewer 2 Report
The research presented in this manuscript is well designed and sufficient discussion on its results. However, to make this study more comprehensive, I suggest authors to consider the following comments.
Comments:
- The Introduction has three paragraphs with only 15 references. Some suggestions on how to improve the citation level are given as, line number (s) 26-28 The statement need several references
- Tablet porosity and tablet tensile strength was studied (Figure 5 and 6, respectively). However, authors did not mention the equation to calculate this parameter in the Method section. It could be more informative to the reader.
- Drug release study was performed up to 21 h (Figure 2), but the authors has selected the time point up to 8 h. It needs explanation.
- In this study, microcrystalline cellulose (MCC) and Lactose monohydrate (LM) was used as filler and its composition in the formulation was dependent with the composition of Hypermellose used in the formulation (Table 1). Obviously, an increased in the HPMC concentration in the formulation decreases the release rate of the drug. However, it is also important to investigate the impacts of change in the composition of MCC and LM in the drug release rate. This is important because, MCC is water insoluble polymer whereas LM is water soluble polymer. Therefore, I request authors to consider this point.
- In the study, authors used different composition of the HMPC in the formulation, HPMC is the non-ionic water soluble polymer and change in the composition of HMPC in the tablet may alter the drug release kinetics. In my opinion, the drug release kinetics need to be analyzed as it is important parameter in matrix based tablet
